# PEDOT:PSS-Coated Polybenzimidazole Electroconductive Nanofibers for Biomedical Applications

**DOI:** 10.3390/polym13162786

**Published:** 2021-08-19

**Authors:** Laura Sordini, João C. Silva, Fábio F. F. Garrudo, Carlos A. V. Rodrigues, Ana C. Marques, Robert J. Linhardt, Joaquim M. S. Cabral, Jorge Morgado, Frederico Castelo Ferreira

**Affiliations:** 1iBB—Institute for Bioengineering and Biosciences and Department of Bioengineering, Instituto Superior Técnico, Universidade de Lisboa, Av. Rovisco Pais, 1049-001 Lisboa, Portugal; laura.sordini@tecnico.ulisboa.pt (L.S.); joao.f.da.silva@tecnico.ulisboa.pt (J.C.S.); fabio.garrudo@tecnico.ulisboa.pt (F.F.F.G.); carlos.rodrigues@tecnico.ulisboa.pt (C.A.V.R.); joaquim.cabral@tecnico.ulisboa.pt (J.M.S.C.); 2Associate Laboratory i4HB—Institute for Health and Bioeconomy, Instituto Superior Técnico, Universidade de Lisboa, Av. Rovisco Pais, 1049-001 Lisboa, Portugal; 3Instituto de Telecomunicações and Department of Bioengineering, Instituto Superior Técnico, Universidade de Lisboa, Av. Rovisco Pais, 1049-001 Lisboa, Portugal; 4CDRSP—Centre for Rapid and Sustainable Product Development, Polytechnic Institute of Leiria, Rua de Portugal-Zona Industrial, 2430-028 Marinha Grande, Portugal; 5Center for Biotechnology & Interdisciplinary Studies, Department of Chemistry & Chemical Biology, Rensselaer Polytechnic Institute, Troy, NY 12180, USA; linhar@rpi.edu; 6CERENA, Department of Chemical Engineering, Instituto Superior Técnico, Universidade de Lisboa, Av. Rovisco Pais, 1049-001 Lisboa, Portugal; ana.marques@tecnico.ulisboa.pt

**Keywords:** electroconductive, nanofibers, electrospinning, PBI, PEDOT:PSS, mesenchymal stem cells

## Abstract

Bioelectricity drives several processes in the human body. The development of new materials that can deliver electrical stimuli is gaining increasing attention in the field of tissue engineering. In this work, novel, highly electrically conductive nanofibers made of poly [2,2′-m-(phenylene)-5,5′-bibenzimidazole] (PBI) have been manufactured by electrospinning and then coated with cross-linked poly (3,4-ethylenedioxythiophene) doped with poly (styrene sulfonic acid) (PEDOT:PSS) by spin coating or dip coating. These scaffolds have been characterized by scanning electron microscopy (SEM) imaging and attenuated total reflectance Fourier-transform infrared (ATR-FTIR) spectroscopy. The electrical conductivity was measured by the four-probe method at values of 28.3 S·m^−1^ for spin coated fibers and 147 S·m^−1^ for dip coated samples, which correspond, respectively, to an increase of about 10^5^ and 10^6^ times in relation to the electrical conductivity of PBI fibers. Human bone marrow-derived mesenchymal stromal cells (hBM-MSCs) cultured on the produced scaffolds for one week showed high viability, typical morphology and proliferative capacity, as demonstrated by calcein fluorescence staining, 4′,6-diamidino-2-phenylindole (DAPI)/Phalloidin staining and MTT [3-(4,5-dimethylthiazol-2-yl)-2,5 diphenyl tetrazolium bromide] assay. Therefore, all fiber samples demonstrated biocompatibility. Overall, our findings highlight the great potential of PEDOT:PSS-coated PBI electrospun scaffolds for a wide variety of biomedical applications, including their use as reliable in vitro models to study pathologies and the development of strategies for the regeneration of electroactive tissues or in the design of new electrodes for in vivo electrical stimulation protocols.

## 1. Introduction

Bioelectricity drives several biological processes, including cell and tissue growth/development, wound healing and tissue regeneration [1]. Bioelectrical triggered cell signaling affects transcriptional cascades, and thus can influence changes in cell fate, potentially affecting processes such as proliferation, differentiation, migration, morphology and apoptosis [2]. Aside from excitable cells, such as neurons and muscle cells, there are several other cell types that respond to electric fields in the human body, such as fibroblasts [3], osteoblasts [4], chondrocytes [5] and mesenchymal stromal cells (MSCs) [6]. MSCs have been considered a promising cell source for developing novel cell-based therapies, mainly due to their availability from a wide variety of tissues, high proliferative capacity, low-immunogenicity and beneficial immunomodulatory/trophic properties [7]. Regarding tissue regeneration, external electrical stimulation has been shown to promote bone healing both in animal experiments and clinical treatments [8]. Additionally, preclinical and clinical studies have shown superior healing of cartilage defects with MSCs after the application of electrical current [9,10].

Novel electrically conductive biocompatible materials, capable of delivering electrical stimuli to cells have been developed, envisaging new approaches for regenerative medicine. Such materials can be directly implanted for cell stimulation in vivo, used to guide cell differentiation in vitro, or employed as vehicles to deliver biochemical agents when electrically stimulated [11]. As such, when used as tissue substitutes (alone or in combination with electrical stimulation), these materials can potentially facilitate the healing process of diseased or damaged tissues. The most commonly used electrically conductive materials in biomedical applications are metals, carbon nanotubes and conductive polymers [12]. Among these, electrically conductive polymers are the most versatile as they can be easily functionalized for a specific application and can be processed using various manufacturing techniques, such as casting, 3D melt-extrusion, electrospinning and 3D printing [13]. Particularly, conjugated polymers with a π-conjugated backbone can be designed/modified to display high electrical conductivity, outstanding photophysical properties and excellent biocompatibility [14]. The most widely investigated conjugated polymers for biomedical applications include polypyrrole (PPy), polyaniline (PANI), polythiophene (PTh) and its derivatives such as poly (3,4-ethylenedioxythiophene) (PEDOT). When oxidized, PEDOT is usually stabilized with a polyelectrolyte, polystyrene sulfonate (PSS), and the resultant PEDOT:PSS blend presents high electrical conductivities [15,16].

Polymeric nanofibers have great potential as tissue engineering scaffolds, since they can closely mimic the nanoscale structural features of the native extracellular matrix (ECM). Moreover, the high porosity, pore interconnectivity and surface area provided by electrospun scaffolds promote cell adhesion, migration and proliferation and enable efficient nutrient supply and waste removal [17,18]. Electrospun nanofibrous scaffolds have been employed in numerous tissue regeneration applications including skin, muscles, bone, cartilage, nerves and blood vessels [19,20,21]. Electrically conductive nanofibers can provide both topographical and electrical cues, being particularly promising to modulate the behavior of the cells [22]. In particular, both neurogenic and chondrogenic differentiation of human MSCs were reported to be enhanced upon electrical stimulation, when such cells were cultured on electrically conductive scaffolds without chemical cues [23,24]. Prabhakaran et al. presented a comparative study to evaluate nerve stem cell growth with electrical stimulation (ES) on conductive and non-conductive nanofibers and demonstrated that, when using a conducting polymer, high proliferation was obtained on nanofibers with a small diameter and high tensile strength [25]. However, the production of an electrospun nanofiber scaffold with a narrow fiber diameter distribution and morphology still remains challenging. Further studies on nanofibers are needed to obtain effective 3D matrixes for MSC culturing and differentiation and the application of external stimuli (e.g., electrical) to help identify a beneficial tool for use in regenerative medicine.

Previous results from our group have demonstrated the superior properties of water stable films made of PEDOT:PSS CLEVIOS P VP.AI 4083 (1:6 PEDOT:PSS weight ratio) crosslinked with (3-glycidyloxypropyl) trimethoxysilane (GOPS) with electrical conductivity in the range of several tens of S·m^−1^. Such highly conductive films were then successfully used to support neural stem cells differentiation under electrical stimulation [16]. However, processing PEDOT:PSS into robust 3D fibers of nanoscale diameter, without losing its conductivity, is a very challenging task.

In the current study, PEDOT:PSS cross-linked with GOPS was investigated, for the first time, as a coating to increase the conductivity and the biocompatibility of electrospun fibrous scaffolds made of a semi-conjugated polymer, polybenzimidazole (PBI).

PBI is a fully aromatic heterocyclic polymer (Figure 1), chemically stable and resistant to heat and mechanical load. PBI is rarely used in the context of medical applications and therefore it is not an obvious choice as carrier electrospun fibers to be coated with PEDOT:PSS. A recent study by our group has demonstrated, for the first time, a superior biocompatibility property of polybenzimidazole (PBI) electrospun nanofibers to support neural stem cell culture [26]. In our study [26], we were able to increase PBI nanofiber conductivity through doping with different acids. Nonetheless, the highest value obtained was far below 1 S·m^−1^. PBI is easily electrospun to form nanofibers, that can be protonated upon contact with the acidic PEDOT:PSS dispersion (pH ≈ 1–2), potentially contributing to ionic interactions with PEDOT:PSS. Moreover, PBI has another important advantage, enabling the stabilization of the coating with PEDOT:PSS solution. For coating stabilization, the material has to undergo an annealing treatment at 150 °C, a temperature that is higher than the melting temperature of several polymers, but suitable for PBI, which is thermostable up to 400 °C [27].

The aim of this work is the fabrication of novel PBI-based electrospun nanofibrous scaffolds with enhanced electroconductivity through coating with PEDOT:PSS crosslinked with GOPS. Two different coating methods (spin coating vs. dip coating) were used to generate PEDOT:PSS-coated PBI nanofibers. The scaffolds obtained were characterized in terms of their morphology, fiber diameter, electrical conductivity and Fourier-transform infrared (FTIR) spectra. Moreover, the biocompatibility of the developed scaffolds was assessed in vitro using human bone marrow-derived MSCs (hBM-MSCs). hBM-MSC were selected considering their relevance in cell therapy and tissue engineering strategies and the final goal to produce biocompatible nanofibers with adequate electrical conductivity for biomedical applications. To our knowledge, this is the first study combining PEDOT:PSS with PBI electrospun nanofibers to generate biocompatible scaffolds with enhanced electrical conductivity.

## 2. Materials and Methods

### 2.1. Preparation of Material Solutions

Celazole^®^ S26 solution, containing 26 wt % PBI and 1.5 wt % lithium chloride in N,N-dimethylacetamide (DMAc), was purchased from PBI Performance Products, Inc. (Charlotte, NC, USA). This starting solution was diluted to 13% using DMAc (Sigma-Aldrich, St. Louis, MO, USA), and the mixture was left overnight under mechanical stirring to obtain a homogeneous solution. The aqueous PEDOT:PSS-coating dispersion (used for PBI fibers coating) was prepared by addition of ethylene glycol (EG,1:4 EG to PEDOT:PSS volume ratio), dodecylbenzenesulfonic acid (DBSA, 0.5 μL·mL^−1^), and GOPS (10 μL·mL^−1^) to PEDOT:PSS regular dispersion (solid content 1.3–1.7%, CLEVIOS P AI 4083, Heraeus, Germany), which has been filtered using a 0.45 µM filter [16]. DMAc, EG, DBSA, hydrochloric acid (HCl), isopropanol and GOPS were purchased from Sigma-Aldrich, St. Louis, MO, USA and used as received.

### 2.2. Production of Electrospun PBI Nanofibers

PBI 13 wt % solution in DMAc was loaded into a 10 mL Luer lock syringe placed in a syringe pump and connected to a polytetrafluoroethylene (PTFE) tube, which was attached on the other end to a 21G needle. The electrospinning procedure was performed using a vertical set-up and a flat aluminum collector placed at a distance of 11 cm from the needle tip. The electrospinning operating conditions were 20 kV direct current (DC) applied voltage and a controlled flow rate of 0.3 mL·h^−1^. The fibers were electrospun directly on clean glass coverslips placed on the top of the collector. The process was carried out at room temperature (21–24 °C) and with a relative humidity of 50%. Finally, the collected fibers were left to air dry overnight.

### 2.3. Coating of Electrospun PBI Nanofibers with PEDOT:PSS

Two different coating methods, spin coating and dip coating, were tested in this work. The composition of the aqueous PEDOT:PSS coating dispersion, used in both methods and comprising PEDOT:PSS and cross-linking agents, is described in Section 2.1. In the spin coated method the aqueous PEDOT:PSS-coating dispersion was spun (Spin-Coater KW-4A, Chemat Technology, Northridge, CA, USA) on the PBI nanofiber meshes at a spinning speed of 1800 rpm for 60 s. In the dip coated method, the PBI nanofibers were immersed in the aqueous PEDOT:PSS coating dispersion for 24 h, after which the excess of coating solution was allowed to drain out of the sample. In both methods, the coated fibers were annealed at 150 °C for 2 min to obtain the final cross-linked PEDOT:PSS-coated PBI nanofiber meshes.

### 2.4. Scanning Electron Microscopy

The morphology of the electrospun nanofiber mats was evaluated by scanning electron microscopy (SEM, JEOL, JSM-7001F model, Japan) at an accelerating voltage of 20 kV. Prior to imaging, the samples were coated with a 30 nm layer of gold/palladium (Polaron model E5100 sputter coater, Quorum Technologies, UK). The average fiber diameter of the electrospun scaffolds was determined by measuring 40 individual fibers from at least 5 different SEM images (5000X) using ImageJ software (ImageJ 1.51f, National Institutes of Health, Bethesda, MD, USA).

### 2.5. Attenuated Total Reflectance Fourier-Transform Infrared (ATR-FTIR) Spectroscopy

The ATR-FTIR analysis was performed using a Spectrum Two FT-IR Spectrometer (Perkin-Elmer, Waltham, MA, USA). The samples submitted to ATR-FTIR were: (i) pristine electrospun PBI fibers, (ii) cross-linked PEDOT:PSS-spin coated PBI fibers, (iii) cross-linked PEDOT:PSS-dip coated PBI fibers, (iv) non cross-linked PEDOT:PSS pellets, and (v) cross-linked (PEDOT:PSS:GOPS) pellets. Transmittance spectra were obtained over the region from 500 to 4000 cm^−1^, with a resolution of 4 cm^−1^ and an accumulation of 8 scans. Pellets were prepared by simple drop casting allowing the water evaporation at 70 °C (overnight) and labeled as PEDOT:PSS or PEDOT:PSS:GOPS, respectively, when made from regular PEDOT:PSS dispersion or from the aqueous PEDOT:PSS-coating dispersion (see Section 2.3 for composition), respectively. Both pellets were then annealed at 150 °C for 2 min.

### 2.6. Four-Point Probe Electroconductivity Measurements

Four stripes of gold were deposited on the samples by physical vapor deposition (PVD), using an Edwards E306A thermal evaporator, across the entire surface and with equal distance from each other. Electrodes were put in direct contact with the gold stripes. Measurements were performed in triplicate (*n* = 3) and averaged. The resistance (*R*) was estimated according to Ohm’s law (*R* = ΔV/I), correlating the difference of electrical potential measured between the two inner contacts for different values of electrical current applied at the outer contacts. At a constant temperature, the: resistivity (*σ*) and conductivity (*σ*) can be calculated, respectively, from Equations (1) and (2):*R* = *ρ* (*L*/*A*) (S^−1^ or Ω)(1)
*σ* = 1/*ρ*  (S·cm^−1^)(2)
where *L* is the distance between the two inner contacts and *A* is the sample cross-section (i.e., sample thickness times the sample width). For thickness measurements, the sample was placed on a glass support, cut with a scalpel all of the way through to the glass surface, and the depth of the cut was estimated by profilometry (Bruker’s Dektak^®^ 3.21 Profilometer (Bruker, Billerica, MA, USA) upon surface scanning perpendicular to the cut.

### 2.7. Cell Culture

hBM-MSCs were isolated from bone marrow aspirates following a protocol previously developed in our group [28]. The aspirates were obtained from a healthy donor (male, 36 years) upon informed consent, with the approval of the ethics committee of Instituto Português de Oncologia Francisco Gentil. Isolated hBM-MSCs were cultured using low-glucose Dulbecco’s Modified Eagle’s Medium (DMEM, Gibco, ThermoFisher, Waltham, MA, USA) supplemented with 10% *v*/*v* fetal bovine serum MSC-qualified (FBS, Life Technologies, Carlsbad, CA, USA) and 1% *v*/*v* antibiotic–antimycotic (Anti-Anti, ThermoFisher, Waltham, MA, USA), and kept at 37 °C and 5% CO_2_ in humidified atmosphere. All the experiments were performed using cells between passages P4–P6 and the culture medium was fully replaced every 3–4 days.

### 2.8. hBM-MSCs’ Seeding on Electrospun Scaffolds

The three types of electrospun scaffolds prepared in this work (i.e., pristine electrospun PBI nanofibers, cross-linked PEDOT:PSS-spin coated PBI nanofibers and cross-linked PEDOT:PSS-dip coated PBI nanofibers) were placed in ultra-low attachment 24-well culture plates (Corning, NY, USA) and sterilized with 1% Anti-anti solution in PBS (Gibco, ThermoFisher, Waltham, MA, USA) for 24 h. The samples were then washed twice with PBS and left immersed in culture medium DMEM + 10% FBS + 1% Anti-anti for 3 h. Afterwards, the culture medium was removed and scaffolds were seeded with a density of 40,000 hBM-MSCs per scaffold. The cells were left for 2 h at 37 °C and 5% CO_2_ without culture medium to promote initial cell attachment to the scaffolds. Culture medium was fully renewed every 3–4 days.

### 2.9. Evaluation of the Viability and Proliferation of hBM-MSCs on PEDOT:PSS-Coated PBI Electrospun Scaffolds

The viability and proliferation of hBM-MSCs on the three scaffolds prepared (pristine electrospun PBI fibers, cross-linked PEDOT:PSS-spin coated PBI fibers and cross-linked PEDOT:PSS-dip coated PBI fibers) were evaluated at days 1 and 7 by assessing cell metabolic activity using the MTT (3-(4,5-dimethylthiazol-2-yl)-2-5 diphenyl tetrazolium bromide, MTT Cell Growth Assay Kit, Sigma-Aldrich, St. Louis, MO, USA) assay following the manufacturer’s guidelines. Briefly, after washing the samples with PBS, the cells were incubated with MTT solution (1 mg·mL^−1^, prepared in PBS) and incubated for 4 h at 37 °C as previously described [29,30]. The resulting formazan salt was then dissolved using a solution of 0.1 M HCl in isopropanol under agitation for 5 min, and the absorbance values of the resultant solutions were measured using a plate reader (Infinite M200 PRO, TECAN, Switzerland) at 570 nm. Three scaffolds (*n* = 3) were used for each condition and the absorbance values were measured in triplicate. Acellular scaffolds (i.e., scaffolds that have not been seeded with cells) were used as blank controls.

The viability of hBM-MSCs on the scaffolds was also confirmed by calcein staining at days 1 and 7. For that, cells were washed once with PBS and incubated with a calcein staining solution (4 mM in PBS, ThermoFisher, Waltham, MA, USA) for 40 min at 37 °C. Afterwards, the cells were washed twice with PBS and imaged using a fluorescence microscope (LEICA DMI3000B, Leica Microsystems, Germany) equipped with a digital camera (Nikon DXM1200F, Nikon Instruments Inc., Japan).

### 2.10. Assessment of hBM-MSCs’ Morphology on PEDOT:PSS-Coated PBI Electrospun Scaffolds

To evaluate the morphology of the cells on the electrospun scaffolds, DAPI/Phalloidin staining was performed at days 1 and 7. Cells were fixed for 30 min with 4% (*v*/*v*) paraformaldehyde (PFA, Santa Cruz Biotechnology, Dallas, TX, USA) solution (in PBS) and permeabilized with 0.1% Triton X-100 (Sigma-Aldrich, St. Louis, MO, USA) for 10 min. Then, the samples were incubated with Phalloidin-TRITC (5 μg·mL^−1^, Sigma-Aldrich, St. Louis, MO, USA) for 45 min in the dark, washed twice with PBS, and counterstained with DAPI (1.5 μg·mL^−1^, Sigma-Aldrich, St. Louis, MO, USA) for 5 min in the dark. Finally, the cells were washed again twice with PBS and the staining was observed in a fluorescence microscope (LEICA DMI3000B, Leica Microsystems, Germany) equipped with a digital camera (Nikon DXM1200F, Nikon Instruments Inc., Japan).

## 3. Results and Discussion

### 3.1. Morphological Characterization of Electrospun Scaffolds

In this work, we tested two different methods, spin coating and dip coating, to coat the electrospun PBI fibers with PEDOT:PSS aiming to increase the electroconductivity of the obtained fiber mat. After PEDOT:PSS coating, followed by thermal annealing, a porous nanoweb was obtained. The morphology of the electrospun fibers mats was analyzed by SEM (Figure 1). The fibers of neat PBI are randomly oriented, homogeneous, showing few defects, and have an average diameter of 184 ± 59 nm. This average diameter value is similar to that obtained by Jahangiri et al. [31], but slightly larger than the values obtained in previous works [26,32], which we attribute to the milder electrospinning conditions used in the present study, namely the lower voltage and tip-to-collector distance. After coating, the diameter of the fibers increased to 256 ± 59 nm and 279 ± 54 nm, for spin coated and dip coated fibers, respectively. The increase in fiber diameter suggests that both coating methods are effective to modify the PBI surface. The dip coated samples exhibit a slightly higher diameter than for the spin coating ones, which reflects the adsorption of more material on the fibers surface, in agreement with previously reported studies [33].

The diameters of the obtained fibers are within the nanometer range, all below 300 nm. The fiber diameter size is a very important factor for tissue engineering applications employing MSCs. For example, Lü and colleagues [34] compared poly(3-hydroxybutyrate-co-3-hydroxyvalerate) (PHBV) electrospun fibers with diameters in the micrometer (2110 nm) and nanometer (383 or 600 nm) range and observed that nanometer size fibers promote higher attachment and proliferation of BM-MSCs in the scaffold, which is important for the success of tissue engineering strategies. A similar trend was also observed by Jia and colleagues [35] for BM-MSCs cultured on poly (L-lactic acid)/collagen (PLLA/Coll) nanofibers targeting vascular tissue regeneration, and by Yang and colleagues [36] for neural stem cells cultured on PLLA nano/micro fibrous scaffolds.

### 3.2. ATR-FTIR Analysis

The ATR-FTIR spectra of the materials are shown in Figure 2. ATR-FTIR spectrum of pristine electrospun PBI fibers (Figure 2A a) displays the characteristic peaks of PBI structure corresponding to substituted benzene rings (out-of-plane bend vibrations for the C–H bonds at 691 cm^−1^ and 799 cm^−1^), to imidazole rings (breathing mode at 1289 cm^−1^) and benzimidazole rings (in-plane deformation at 1444 cm^−1^). The peak at 1618 cm^−1^ corresponds to C = N stretching, also present in the imidazole groups. The region between 2400 and 3955 cm^−1^ is attributed to amine groups, widely distributed through the polymer’s structure [26,37]. The presence of the solvent used for electrospinning, DMAc, was not detected.

As a first approach, we normalized the peak intensities of all the PBI-containing fibers to the common peak at 799 cm^−1^ (Figure 2A). A further analysis based on normalization of spectra to the common at peak 1443 cm^−1^ can be found on Appendix A (Appendix A) allowing further detailed discussion on PEDOT:PSS andGOPS contribution to the spectra.

The spectra of the cross-linked PEDOT:PSS-coated PBI fiber samples obtained by spin and dip coating are depicted in Figure 2A b,A c, respectively. They combine the characteristic peaks of the starting materials, PBI and PEDOT:PSS with GOPS, confirming that the coating was successful for both spin and dip coated samples. Thus, peaks at 799 cm^−1^, 1298 cm^−1^, 1444 cm^−1^ and 1632 cm^−1^ are attributed to PBI, and peaks at 1034 cm^−1^, 1083 cm^−1^ and the double peaks at 2855/2924 cm^−1^ are attributed to PEDOT:PSS cross-linked with GOPS. The region between 2400 and 3955 cm^−1^ arises from an overlap of the corresponding regions of PBI and PEDOT:PSS with GOPS. This region is associated with the hydroxyl and amine groups present in either material, and ultimately with residual free water present/absorbed by the samples [38,39]. The broad peak at 1643 cm^−1^ from PEDOT:PSS with GOPS slightly overlaps the sharp 1444 cm^−1^ peak of imidazole/benzimidazole rings of PBI in both dip coated and spin coated samples. The overlay seen of both PBI and PEDOT peaks in our PEDOT:PSS-coated PBI fibers is consistent with the phase separation in both the dip-coated and spin-coated fibers and confirms that the coating was successful.

In the PEDOT:PSS-dip coated PBI fibers (Figure 2A c), the double peak 2855/2924 cm^−1^ is more pronounced than the PEDOT:PSS-spin coated PBI fibers (Figure 2A b). In the corresponding 3000–3660 cm^−1^ region, we expect peaks arising not only from both amine (PBI) and hydroxyl (water) groups present in either PBI or PEDOT:PSS, but also from residual free water. In the case of PEDOT:PSS-dip coated PBI fibers, the relative intensity of this area is smaller when compared to PEDOT:PSS-spin coated PBI fibers.

To perform a better analysis of the behavior of PEDOT:PSS in all our samples without the bias resulting from PBI, we normalized the peak intensities of all PEDOT-containing samples to the common peak at 2324 cm^−1^ (Figure 2B). PEDOT:PSS spectra (Figure 2B x) have two peaks at 665 cm^−1^ and 860 cm^−1^ (thiophene ring). There are also two double peaks at 998 and 1000 cm^−1^ (C–S bond in thiophene ring) and at 1123 and 1158 cm^−1^ (C–O stretching in PEDOT and sulfoxide in PSS). A broad peak is also seen at 1595 cm^−1^ (thiophene ring) and is also present in the spectra of PEDOT:PSS with GOPS (Figure 2B y). Moreover, a peak common to all PEDOT-derived compounds tested in this work is seen at 2325 cm^−1^ and next to it there is a broad peak at 2656 cm^−1^ (alkane C–H bonds) [40,41]. The spectrum of the PEDOT:PSS pellet with GOPS (Figure 2B y) has a small peak at 663 cm^−1^ (C–S stretching in the thiophene ring), two major peaks at 1007/1028 and 1092 cm^−1^ (sulfoxide groups in PSS), and a small peak at 1194 cm^−1^ (sulfonate group present in the GOPS cross-linking). At 1643 cm^−1^, there is a broad peak associated with the C = C bond in the thiophene ring. A double peak at 2855/2924 cm^−1^ (C–H bond stretching and aldehyde bonds arising from GOPS cross-linking [42,43]) and a broader peak at 3357 cm^−1^ (2990–3640 cm^−1^ region) (intermolecular hydrogen bonds) can also be seen.

In the original PEDOT:PSS samples, the peak intensity of this area drastically increases with GOPS cross-linking. This is mainly due to an increase in the number of sulfoxide bonds (1007/1028 cm^−1^ and 1092 cm^−1^) established between PSS and GOPS, leading also to a decrease in the number of sulfonate (1197 and 1408 cm^−1^) groups present. A first look at the 750–1250 cm^−1^ region (thiophene ring) evidences that the overall peak intensity is higher for dip coated samples than for the spin coated samples.

In the PEDOT:PSS-dip coated PBI fibers (Figure 2B c’), the intensity of the sulfonate peak at 1197 cm^−1^ is smaller than the spin coated sample (Figure 2B b’). This suggests the higher stability of the PEDOT:PSS with GOPS layers formed around the PBI fiber, since a greater amount of more stable bonds is present. Another indication of a more successful cross-linking of PEDOT:PSS in the dip coated sample is a more defined double peak at 2855/2924 cm^−1^, which is present in the original PEDOT:PSS cross-linked with GOPS.

Both samples were cross-linked in similar conditions (150 °C, 2 min) and similar fiber diameters were obtained. We hypothesize that the longer incubation of the dip coating method is a more efficient method to deposit a stable/cross-linked PEDOT:PSS layer on the PBI fibers. A consequence of a more stable PEDOT layer is a better overall electroconductivity of the obtained PEDOT:PSS-coated PBI fibers obtained by dip coating.

The results in Appendix A further suggest a higher amount of PEDOT:PSS on fiber samples obtained by dip coating than on the ones obtained by spin coating. Both the ATR-FTIR intensity of normalized peaks (Appendix A) and the TGA mass loss percentage (Appendix A and Appendix A), respectively, at wavelengths and temperatures allocated to PEDOT:PSS, are higher for fibers samples obtained by dip coating than for the ones obtained by spin coating.

### 3.3. Electroconductivity Measurements

The electrical conductivity of the electrospun scaffolds was measured using the four-probe method. PBI nanofibers displayed a conductivity of 3.0 × 10^−5^ S·m^−1^, which is consistent with values obtained previously by our group [26] and with values reported by others [44]. 

The PEDOT:PSS cross-linking solution used in this work was previously tested by Pires et al. in the form of films, using slightly different annealing conditions, and showed a conductivity of 5.8 S·m^−1^ [16]. The conductivity obtained for a cross-linked PEDOT:PSS film deposited on glass using the new annealing conditions (150 °C, 2 min) is 500 S·m^−1^, which is consistent with our most recent studies [45].

After coating with PEDOT:PSS, the conductivity of the fiber mats was determined to be 28.3 S·m^−1^ for spin coated scaffolds and 147 S·m^−1^ for dip coated ones, where the thickness used in the electrical conductivity calculations was taken as the total thickness of the materials (PBI fiber mat with PEDOT:PSS-coating layer). As these scaffolds behave as a two layer-materials, with the PBI, having a lower electrical conductivity, as a substrate, the electrical conductivity is mainly that of the PEDOT:PSS coating layer. Therefore, the intrinsic electrical conductivity of PEDOT:PSS layer, thinner than the entire scaffold, will be higher than the calculated values for the PEDOT:PSS-coated scaffolds and should approach the value of 500 S·m^−1^ mentioned above. 

Accordingly, we attribute the higher conductivity values obtained for the dip coated samples in comparison with that of the spin coated sample mainly to the higher thickness of the PEDOT:PSS coating layer present, as suggested by the fiber diameter size and ATR-FTIR studies in Section 3.1 and Section 3.2, respectively. However, possible differences in the intrinsic electrical conductivity of the cross-linked PEDOT:PSS deposited by dip and spin coating, as a result of differences in composition, cross-linking degree and phase separation induced by the different deposition and drying conditions cannot be ruled out.

Table 1 compares the characteristics of the obtained PEDOT:PSS-coated PBI fibers with other previously reported studies on electrically conductive electrospun fibers containing PEDOT:PSS. From the comparative analysis of the studies, we may conclude that our PEDOT:PSS-coated PBI scaffolds present an interestingly high electrical conductivity and small fiber diameter. Many studies in the literature have shown that a reduction in the fiber diameter of electrospun scaffolds significantly increases cell attachment and growth, due to an increase in surface area and local curvature which results in stronger bonds between cell surface receptors and fibers [46,47]. In particular, Abedi et al. [48] have demonstrated that a lower average fiber diameter have enhanced MSCs’ metabolism and proliferation on electrospun chitosan/PEDOT:PSS electrically conductive scaffolds [48]. 

In a previous study, culturing hMSCs on electrospun nanofibers meshes, Chen et al. fabricated bacterial cellulose nanofibers coated with PEDOT by in situ interfacial polymerization and achieved electrical conductivities of 0.1–10 S·m^−1^ [49]. Despite the differences in the coating methods used, the lower electrical conductivities obtained, in comparison to our study, might be explained by a possible difference in the amount of conductive PEDOT phase within the samples and its oxidative degree, combined with a poor electrical conductivity of about 10^−6^ S·m^−1^ of the supporting bacterial cellulose fibers, which is lower than the values reported for PBI [26]

**Table 1 polymers-13-02786-t001:** Comparison of the electrical conductivity and average fiber diameter values between the nanofibers studied in this work and other PEDOT-containing nanofibers previously reported in the literature. Abbreviations: BC—bacterial cellulose; PA—polyacrylamide; PVA—polyvinyl alcohol; PVC—polyvinyl chloride; PVP—polyvinyl pyrrolidone; n.a.—Not applicable.

Scaffold Material	PEDOT Coating Method	Diameter Range (nm)	Electrical Conductivity *σ* (S·m^−1^)	Biocompatibility (Cell Source/Methods Used)	Main Outcomes	Ref
PBI nanofibers	None	90–325	3.0 × 10^−5^	hBM-MSCs/ MTT assay and fluorescence imaging	Electrospun nanofibers with small diameter and few defects Low electrical conductivity High MSCs viability and proliferation	this work
PEDOT:PSS-coated PBI nanofibers	Spin coating with PEDOT:PSS	150–415	28	hBM-MSCs/ MTT assay and fluorescence imaging	Homogeneous coating of PEDOT:PSS on PBI electrospun nanofibers, GOPS cross-linking Small diameter and high electrical conductivity High MSCs viability and proliferation	this work
PEDOT:PSS-coated PBI nanofibers	Dip coating with PEDOT:PSS	200–430	147	hBM-MSCs/ MTT assay and fluorescence imaging	Homogeneous coating of PEDOT:PSS on PBI electrospun nanofibers, GOPS cross-linking Small diameter and high electrical conductivity High MSCs viability and proliferation	this work
Cellulose/ PEDOT:PSS nanofibers	PEDOT:PSS casting	n.a.	258	n.a.	Homogeneous PEDOT:PSS-coating Good mechanical properties Good electrical and electrochemical properties for application in energy storage devices	[50]
Plasma-modified chitosan/PEDOT/PVA nanofibers	None	170–200	0.1	n.a.	Superior antibacterial activity Small fiber diameter Electroactive properties	[51]
PEDOT/ Chitosan coaxial nanofibers	Non-in situ permeation	500–600	19	Brain neuroglioma cells (BNCs)/MTT assay; fluorescence and SEM imaging; neurite length measurements; (with and without electrical stimulation)	High electrochemical stability and electrical conductivity and ultrasensitive piezoelectric property Promoted adhesion, proliferation, growth, and development of BNCs (increased number and density of axons) under external electrical stimulation (pulse voltage of 400 mV/cm)	[52]
PEDOT:PSS/ PA nanoweb of entangled nanofibers	-	n.a.	311	n.a.	Successful manufacture of soft conductors (conductive materials with an inherent compliance) Low electrical strain sensitivity under large deformation conditions Environmental stability in water (excellent swelling resistance and maintenance of electrical and mechanical properties)	[53]
PEDOT/PVC nanofibers	In situ interfacial polymerization of EDOT	310–1100	780	Human cancer stem cells (hCSCs)/SEM and fluorescence imaging; cell proliferation (DNA content)	Superior mechanical properties and flexibility (almost restored to original shape after serious twisting and crimping) High electrical conductivity Good biocompatibility (3 days of culture)	[54]
Silk fibroin/PEDOT nanofibers	Vapor-phase polymerization of EDOT	510–590	4000	Human unrestricted somatic stem cells (hUSSCs)/MTT assay, fluorescence imaging, H&E staining and gene expression (RT-qPCR)	Ligament-like scaffold: construct of Silk fibroin/PEDOT bilayer nanofibrous scaffold to mimic the aligned collagen fiber bundles and Chitosan sponge coating to mimic the glycosaminoglycans of ECM sheath Electrical stimulation (DC electric pulses) facilitates cell adhesion, proliferation and the expression of genes involved in the healing process (collagen I, collagen III, decorin, biglycan and aggrecan)	[55]
BC/PEDOT:PSS nanofibers	In situ interfacial polymerization of EDOT	30–200	0.1–10	hMSCs/Fluorescence and SEM imaging; MTT assay	Fabrication of 3D conductive nanomaterial with flexibility and mechanical robustness Biocompatible for hMSC culture (with moderate PSS doping concentration below 0.05 M)	[49]

### 3.4. Cell Viability and Proliferation on PEDOT:PSS-Coated PBI Electrospun Scaffolds

In order to assess the biocompatibility of the electrospun scaffolds produced, hBM-MSCs were cultured for 7 days on the surface of the three scaffold types prepared: pristine electrospun PBI fibers, cross-linked PEDOT:PSS-spin coated PBI fibers and cross-linked PEDOT:PSS-dip coated PBI fibers. The results of the MTT assay performed at days 1 and 7 post-seeding are shown in Figure 3. The data obtained show an increase in the number of cells for all the samples over the 7-day period, including the control glass coverslip samples. While at day 1 the cells seeded on cross-linked PEDOT:PSS-spin coated PBI fibers and PEDOT:PSS-dip coated PBI fibers showed a lower conversion of formazan with respect to the ones seeded on pristine PBI fibers, at day 7 the metabolic activity of cells cultured on PEDOT:PSS-coated scaffolds was similar to the ones on the non-coated PBI scaffolds. Accordingly, the fold increase in cell number from day 1 to day 7 was identical in the cross-linked PEDOT:PSS-coated scaffolds (1.6 and 1.7 for spin and dip coated, respectively) and slightly higher than the one for non-coated fibers (1.2). The results at day 7 suggest the presence of a higher (but not statistically significant) number of cells in the PEDOT:PSS-spin coated scaffolds in comparison to the other electrospun scaffolds assessed. This behavior can be explained by the difference in the surface wettability of the scaffolds assessed by contact angle analysis. The contact angle analysis showed that PEDOT:PSS coating increases the hydrophobicity of the PBI fibers, with PEDOT:PSS-spin coated samples more hydrophilic than the PEDOT:PSS-dip coated ones. (see Appendix A in Appendix A).

To the best of our knowledge, this is the first study combining MSCs with PBI-based electrospun fibers. Nevertheless, other studies have demonstrated the biocompatibility of PEDOT-containing nanofibers for MSCs [48,49]. Abedi et al. fabricated and characterized chitosan/PEDOT:PSS nanofibers, which were found to be highly biocompatible when cultured with rat BM-MSCs [48]. Additionally, Chen et al. reported high hMSCs viability when cultured for 7 days on bacterial cellulose/PEDOT:PSS nanofibers with moderate PSS doping concentrations below 0.05 M [49].

hBM-MSCs’ viability and proliferation on the electrospun scaffolds were further confirmed through calcein staining at day 1 and day 7 (Figure 4). The results are in accordance with the MTT assay, as the cells showed high viability and were able to grow on all the tested samples. Particularly, on pristine PBI and cross-linked PEDOT:PSS-spin coated PBI electrospun scaffolds, cells proliferated until reaching almost confluence, whereas in the case of cross-linked PEDOT:PSS-dip coated scaffolds, a lower cell proliferation was observed, possibly due to its higher hydrophobicity (see Appendix A in Appendix A). Accordingly, a study performed by Birhanu et al. demonstrated that adipose-derived MSCs presented a significantly lower proliferative capacity on hydrophobic poly (L-lactic acid) (PLLA) nanofibers in comparison to plasma-treated hydrophilic PLLA scaffolds [56].

### 3.5. Evaluation of hBM-MSCs’ Morphology on PEDOT:PSS-Coated PBI Electrospun Scaffolds

The morphology of the cells cultured on pristine electrospun PBI, cross-linked PEDOT:PSS-spin coated PBI fibers and cross-linked PEDOT:PSS-dip coated PBI fibers was assessed by DAPI/Phalloidin staining at days 1 and 7. Fluorescence images of the DAPI/Phalloidin staining are shown in Figure 5. The results of DAPI/Phalloidin staining are in accordance with the results presented in Figure 3 and Figure 4. As it is possible to observe in Figure 5, the number of hBM-MSCs increased from day 1 to day 7 on PBI fibers and on cross-linked PEDOT:PSS-spin coated PBI fibers. In the cross-linked PEDOT:PSS-dip coated PBI samples, the cell proliferation levels were lower in comparison to the other scaffold conditions. Despite these differences, in all the scaffold conditions, cells were able to spread and display the typical elongated morphology of undifferentiated MSCs, with well-developed actin cytoskeleton. No particular direction of elongation can be determined, which it is consistent with the random nature of the nanofibers mat. Chen et al. had similar results on bacterial cellulose nanofibers coated with PEDOT:PSS. In their study, after 7 days of culture, the MSCs stretched their morphology on the nanofibers and proliferated on all PEDOT-coated samples [49].

In summary, none of the manufactured nanofibers meshes displayed any sign of cytotoxicity and they were all able to support hBM-MSCs proliferation, while providing, at the same time, an advantageous 3D ECM-like environment for cells. Although both coating methods resulted in similar fold increase in cell number over the 7 days culture period, cross-linked PEDOT:PSS-spin coated fibers promoted higher levels of cell adhesion and higher final cell numbers. However, cross-linked PEDOT:PSS-dip coated fibers are significantly more electrically conductive, which may be beneficial for some applications. As future work, we intend to assess the produced PEDOT:PSS-coated PBI fibers coupled with an external electrical stimulation envisaging bone and cartilage tissue engineering applications. For example, Zhu et al. [57] have demonstrated that the proliferation and osteogenic differentiation of hMSCs were enhanced by the application of an electrical stimulation on conductive nanofibers made of poly (L-lactic acid) (PLLA) containing multi-walled carbon nanotubes (MWCNTs). Moreover, hMSCs were able to elongate in the direction of the electrical field. Accordingly, we expect that electrospun scaffolds containing both nanoscale features and electrical conductivity will be highly effective in actively modulating cell functions.

## 4. Conclusions

In this work, we developed and characterized a novel electrical conductive and biocompatible scaffold composed of PEDOT:PSS-coated PBI nanofibers. PBI nanofibers were efficiently produced by electrospinning and coated with PEDOT:PSS cross-linked with GOPS to enhance their electrical conductivity. Two different coating methods were compared: spin coating and dip coating. The obtained electrospun scaffolds showed increased fiber diameters, presenting an average of 256 ± 59 nm for the spin coated PBI fibers and 279 ± 54 nm for the dip coated PBI fibers. Electrical conductivity measurements demonstrated that the coating highly increased the scaffolds’ electrical conductivity, with respect to the pristine PBI fiber mats, reaching 28.3 S·m^−1^ for PEDOT:PSS-spin coated and 147 S·m^−1^ for dip coated electrospun scaffolds, respectively. Additionally, all the developed scaffolds were able to support hBM-MSCs’ adhesion, viability and proliferation. In particular, PBI nanofibers spin coated with cross-linked PEDOT:PSS displayed both electrical conductivity and high biocompatibility. Therefore, PEDOT:PSS-spin coated PBI scaffolds are considered to be the best candidates for further studies envisaging the development of novel tissue engineering substitutes, reliable in vitro models or new electrodes for in vivo electrical stimulation protocols.

## Figures and Tables

**Figure 1 polymers-13-02786-f001:**
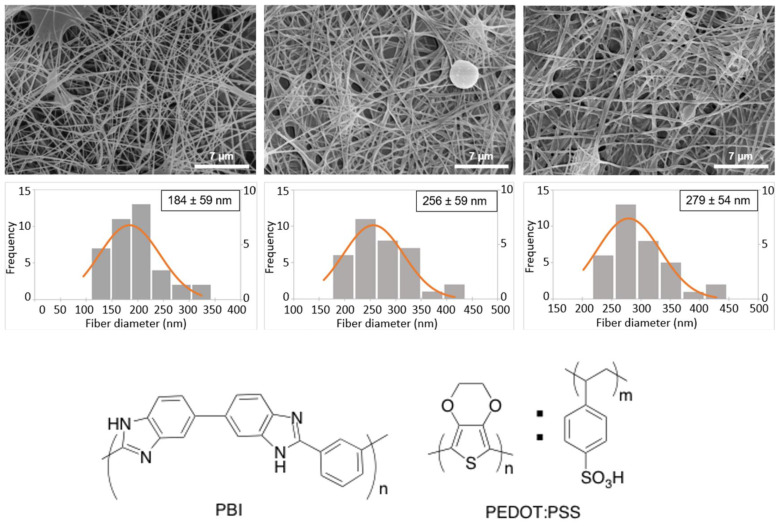
SEM images (**top**) and respective fiber diameter distributions (**middle**) of pristine electrospun PBI, cross-linked PEDOT:PSS-spin coated PBI and cross-linked PEDOT:PSS-dip coated PBI nanofibers. Scale bar: 7 μm. The discreet histogram values are represented on the left *yy*-axis and the normal probability density function each mean and standard deviation are represented on the right *yy*-axis. Chemical structures of poly [2,2′-m-(phenylene)-5,5′-bibenzimidazole] (PBI) and poly (3,4-ethylenedioxythiophene) doped with poly (styrene sulfonic acid) (PEDOT:PSS) are shown in the (**bottom**) panel.

**Figure 2 polymers-13-02786-f002:**
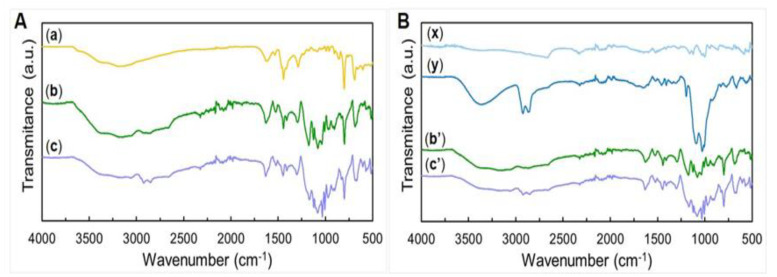
ATR-FTIR spectra of the various samples analyzed. (**A**) Spectra normalized to peak 799 cm^−1^ of (a) PBI electrospun fibers, and PEDOT:PSS/PBI obtained by (b) spin coating and (c) dip coating. (**B**) Spectra normalized to peak 2324 cm^−1^ of PEDOT:PSS/PBI samples obtained by (b’) spin coating and (c’) dip coating, (x) PEDOT:PSS pellet and (y) PEDOT:PSS pellet with GOPS.

**Figure 3 polymers-13-02786-f003:**
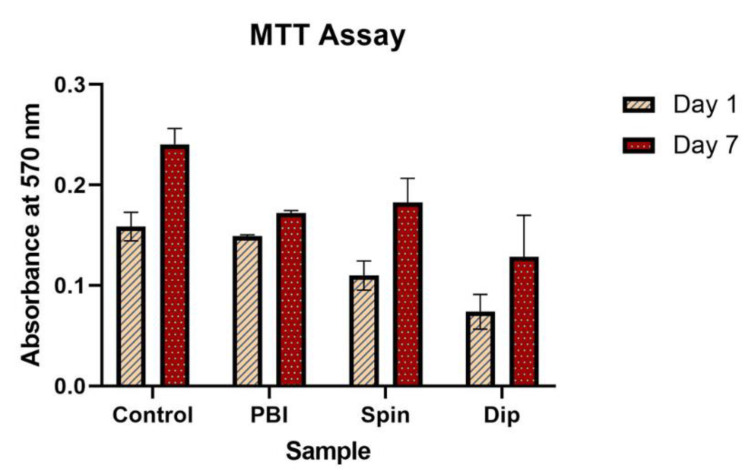
MTT assay results performed after 1 and 7 days of hBM-MSCs culture on the three types of scaffolds: pristine electrospun PBI fibers, PEDOT:PSS-spin coated PBI fibers and PEDOT:PSS-dip coated PBI fibers. Tissue culture plates were used as controls. Results are expressed as mean ± standard deviation (*n* = 3 independent samples).

**Figure 4 polymers-13-02786-f004:**
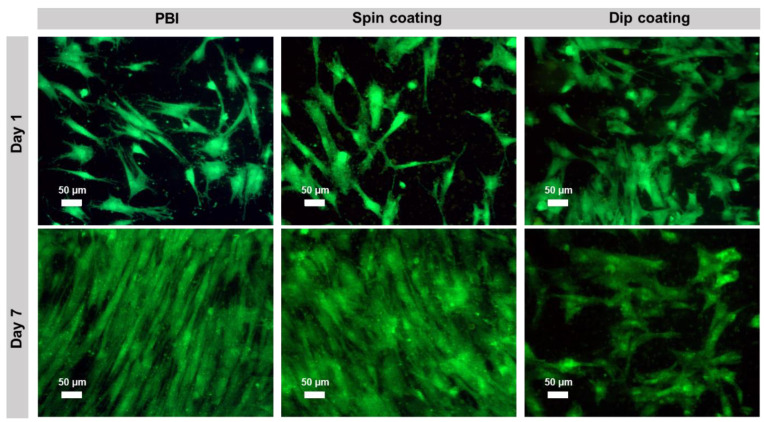
Calcein fluorescence staining of hBM-MSCs cultured on pristine electrospun PBI, PEDOT:PSS-spin coated PBI fibers and PEDOT:PSS-dip coated PBI fibers at days 1 and 7. Viable cells are stained in green. Scale bar: 50 μm.

**Figure 5 polymers-13-02786-f005:**
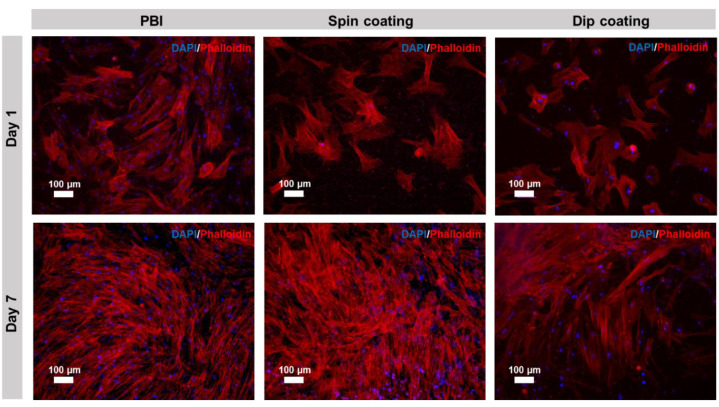
DAPI/Phalloidin immunofluorescence staining of hBM-MSCs cultured on pristine electrospun PBI, PEDOT:PSS-spin coated PBI fibers and PEDOT:PSS-dip coated PBI fibers for days 1 and 7. DAPI stains cell nuclei blue, while Phalloidin stains actin-rich cytoskeleton red. Scale bar: 100 μm.

## Data Availability

Data available on request.

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
