# Peer review of "PEDOT:PSS-Coated Polybenzimidazole Electroconductive Nanofibers for Biomedical Applications"

_polymers, 2021, doi:10.3390/polym13162786_

Round 1

Reviewer 1 Report

The manuscript of Sordini et al., titled ‘PEDOT:PSS-coated polybenzimidazole electroconductive nanofibers for biomedical applications,’ describes the preparation of PEDOT:PSS-GOPS coated PBI by two different methods, describing the electrical properties and biocompatibility of the material.

The work is well organized, and I recommend that the contribution can be published after minor corrections and questions that raised, followed by corrections and suggestions that will improve the work:

  • In the abstract and introduction, there is a focus on PEDOT:PSS coated PBI electrodes. However, when you discuss the ATR-FTIR or the preparation of the materials, you described them as PEDOT:PSS-GOPS, making it confusing because in the middle of the text, I figure out GOPS was used, an not only the PEDOT:PSS with PBI.

For example, in line 74, you claim the following without mentioning that in this work, you will crosslink using GOPS:

One of the disadvantages of PEDOT:PSS is the difficulty to be processed in form of 3D fibers without losing its conductivity. For this reason, in this work, PEDOT:PSS was investigated for the first time as a coating to increase the  conductivity and the biocompatibility of electrospun fibrous scaffolds made of a semi-conjugated polymer, polybenzimidazole (PBI).

In line 101 you repeat the same idea, without mention the crosslinking and the use of GOPS:

The aim of this work was the fabrication of novel PBI-based electrospun nanofibrous scaffolds with enhanced electroconductivity through coating with PEDOT:PSS. Two different coating methods (Spin coating vs. Dip coating) were used to generate PEDOT:PSS coated-PBI nanofibers.

  • In regards to the PEDOT:PSS GOPS films, just by the data shown in this work, you are assuming that there is crosslinking; even though this is pretty known and well documented, references must be added about this. For example, https://doi.org/10.1002/polb.24331 (not necessarily this one).

Corrections and improvements to the text can be found in the attached PDF file.  

Author Response

Detail response to the reviewers

Comments and Suggestions for Authors

Comment 1.

  • The manuscript by Dordini and co-authors reports the production of a new PBI-based electrospun nanofibrous scaffolds with enhanced electroconductivity through coating with PEDOT: PSS (poly(3,4-ethylene dioxythiophene) stabilized with polystyrene sulfonate).

The new scaffold is interesting and the property potential is useful for tissue engineering applications.

However, the biological results must be improved.

The authors have to increase the time of culture of stem cells on both scaffolds. At least, the MTT assay must be performed for 14 days.

The authors have to include the growth curve of stem cells on both scaffolds for 14 days, a least.

The quality of fluorescence images must be improved.

Answer: We thank the reviewer for the constructive comments and for feedback on our work. We are currently preparing the samples and growing the cells to repeat the MTT assay, but expanded to 14 days. 

Note that this additional experiment can not be performed within the requested 10 day deadline. Actually, this experiment will considerably delay the potential publication of the manuscript beyond the 14 days of cell culture, as further time needs to be allocated to sample and cells preparation as well as post-culture characterization. Still, we are now preparing cells and fibers, and if referee and editor think that this extra cell culture time point adds essential value to this study, we will carry on with the experiment. On the other hand, if referee and editor believe that the value of the extra week of cell culture is marginal, we are also happy to interrupt ongoing experimental plans and focus on further studies to be submitted on follow up manuscripts. We will improve the quality of the fluorescence images. Other corrections requested by the referees were carried out.

Reviewer 2 Report

The manuscript by Dordini and co-authors reports the production of a new PBI-based electrospun nanofibrous scaffolds with enhanced electroconductivity through coating with PEDOT: PSS (poly(3,4-ethylene dioxythiophene) stabilized with polystyrene sulfonate).

The new scaffold is interesting and the property potential is useful for tissue engineering applications.

However, the biological results must be improved.

The authors have to increase the time of culture of stem cells on both scaffolds. At least, the MTT assay must be performed for 14 days.

The authors have to include the growth curve of stem cells on both scaffolds for  14 days, a least.
The quality of fluorescence images must be improved.

Author Response

Comments and Suggestions for Authors

Comment 1.

  • The manuscript by Dordini and co-authors reports the production of a new PBI-based electrospun nanofibrous scaffolds with enhanced electroconductivity through coating with PEDOT: PSS (poly(3,4-ethylene dioxythiophene) stabilized with polystyrene sulfonate).

The new scaffold is interesting and the property potential is useful for tissue engineering applications.

However, the biological results must be improved.

The authors have to increase the time of culture of stem cells on both scaffolds. At least, the MTT assay must be performed for 14 days.

The authors have to include the growth curve of stem cells on both scaffolds for 14 days, a least.

The quality of fluorescence images must be improved.

Answer: We thank the reviewer for the constructive comments and for feedback on our work. We are currently preparing the samples and growing the cells to repeat the MTT assay, but expanded to 14 days.

Note that this additional experiment will considerably delay the potential publication of the manuscript beyond the 14 days of cell culture, as further time needs to be allocated to sample and cells preparation as well as post-culture characterization. Still, we are now preparing cells and fibers, and if referee and editor think that this extra cell culture time point adds essential value to this study, we will carry on with the experiment. On the other hand, if referee and editor believe that the value of the extra week of cell culture is marginal, we are also happy to interrupt ongoing experimental plans and focus on further studies to be submitted on follow up manuscripts. We will improve the quality of the fluorescence images. All the other suggestions made by the referees had been implemented.